# Climate Change Impact on Wheat Performance—Effects on Vigour, Plant Traits and Yield from Early and Late Drought Stress in Diverse Lines

**DOI:** 10.3390/ijms23063333

**Published:** 2022-03-19

**Authors:** Yuzhou Lan, Aakash Chawade, Ramune Kuktaite, Eva Johansson

**Affiliations:** Department of Plant Breeding, The Swedish University of Agricultural Sciences, P.O. Box 190, SE-23422 Lomma, Sweden; yuzhou.lan@slu.se (Y.L.); aakash.chawade@slu.se (A.C.); ramune.kuktaite@slu.se (R.K.)

**Keywords:** spring wheat, early vigour, image-based phenotyping, drought, stress-tolerance index, yield performance

## Abstract

Global climate change is threatening wheat productivity; improved yield under drought conditions is urgent. Here, diverse spring-wheat lines (modern, old and wheat-rye introgressions) were examined in an image-based early-vigour assay and a controlled-conditions (Biotron) trial that evaluated 13 traits until maturity. Early root vigour was significantly higher in the old Swedish lines (root length 8.50 cm) and introgressed lines with 1R (11.78 cm) and 1RS (9.91 cm) than in the modern (4.20 cm) and 2R (4.67 cm) lines. No significant correlation was noted between early root and shoot vigour. A higher yield was obtained under early drought stress in the 3R genotypes than in the other genotype groups, while no clear patterns were noted under late drought. Evaluating the top 10% of genotypes in terms of the stress-tolerance index for yield showed that root biomass, grains and spikes per plant were accountable for tolerance to early drought, while 1000-grain weight and flag-leaf area were accountable for tolerance to late drought. Early root vigour was determined as an important focus trait of wheat breeding for tolerance to climate-change-induced drought. The responsible genes for the trait should be searched for in these diverse lines. Additional drought-tolerance traits determined here need further elaboration to identify the responsible genes.

## 1. Introduction

As one of the major staple food sources around the world, wheat provides approximately 20% of the calories and proteins to the daily human diet [1]. According to the latest update in 2020 by the Food and Agriculture Organization of the United Nations (FAO), 219 million ha were harvested, which makes wheat the most widely grown crop; meanwhile, the global production of 761 million tonnes strengthens its position as the world’s second-largest crop (https://www.fao.org/faostat/en/#data/QCL, accessed on 26 February 2022). With the pronounced global climate changes (i.e., rising temperature), prolonged shortages of water supply (drought stress) are becoming increasingly frequent, thereby depleting the ecophysiological performance of plants [2,3]. Therefore, drought has been placed at the top of the environmental stresses due to its severe impact on crop productivity as compared to other natural abiotic stresses [4]. Severe drought-induced yield loss in wheat has been reported in several regions of the world [5,6,7]. Due to the unpredictability of the natural environment, drought stress can threaten a wheat plant at any growth stage throughout its entire life cycle. Late drought has been reported to have a more significant impact on yield loss than early drought [8]. However, in some countries such as Sweden and Denmark, spring wheat is regularly affected by drought during late spring in the early stages of crop development, when the roots have not been fully developed [9]. Therefore, to better understand the mechanism of drought tolerance, the impact of early stress on some phenological and yield-related morphological traits should not be ignored.

Water deficit inhibits the growth of plants by inducing changes of different types, i.e., to the physiological, biochemical, morphological, and molecular characteristics [10]. Many traits besides yield are significantly influenced by drought stress, i.e., flag-leaf area [11], root and plant biomass [12,13], days to anthesis and tillers per plant [14]. To screen for genotypes with high yield potential under stress conditions, a stress-tolerance index (STI) was developed and is used as an effective selection criterion [15]. With the predicted climate change, the need to improve the drought tolerance of wheat has become necessary in many regions of the world.

In addition to traditional labour-intensive trait measurements, scientists have introduced novel sensor-based non-invasive phenotyping techniques in order to investigate plants more efficiently. Various high-throughput phenotyping systems have been established for the study of agronomy traits, i.e., proximal-sensing carts [16], field-scanner systems [17], unmanned aerial systems [18], and automated, standalone systems for controlled growth conditions [19]. Using such systems, the early growth of wheat was found to be correlated to the tolerance to drought conditions, as the tolerant lines tended to display fast early growth [20]. Moderate correlations were identified between early root traits from controlled climatic conditions and drought scores from field trials [13]. 

The commonly used term ‘wheat’ usually refers to bread wheat (*Triticum aestivum* L.) that belongs to the tribe Triticeae and the family Poaceae [21,22]. Over years of domestication and breeding, several types of wheat have been developed for different purposes. Modern cultivars mainly aim for a high and stable yield while old breeding lines, landraces and primitive forms of wheat might contain genes of relevance to sustain the varying climate changes [23,24]. Furthermore, wild relatives and landraces of wheat are being used as a unique source of genetic variation to compensate for the low diversity of modern cultivars [25,26]. The successful transfer of genes from the non-*Triticum* species of rye (*Secale cereale*) have yielded disease-resistant wheat cultivars [27,28,29]. In addition to the disease resistance, the rye chromosome 1RS that was transferred to wheat was reported to carry genes that relate to root biomass [30,31], which could potentially improve tolerance to drought stress.

The present study aimed to use the performance of a broad set of phenotypic traits to characterize drought-stress tolerance in a wide variety of modern, ancient, old, and alien introgressed spring-wheat lines. Furthermore, drought-responding characteristics were related to the genetic background of the material. For successful evaluation, plants were subjected to early or late drought stress in controlled conditions and the performance of the plants was evaluated at the seedling and maturity stage using a combination of classical agronomic traits, including the calculation of STI, and an image-based phenotyping technique. The hypothesis behind this study was that the genes are present in a sufficiently genetically broad wheat material, so that drought-stress tolerance and the genetic background for such tolerance can be identified.

## 2. Results

### 2.1. Early Root and Shoot Development 

ANOVA clearly showed that both digital-root length (DRL) and digital-leaf area (DLA) varied significantly (*p* < 0.001) among the evaluated genotypes (Appendix A), with values ranging from 2.83 cm to 16.13 cm for DRL and from 4.03 cm^2^ to 12.52 cm^2^ for DLA. No significant correlation was found between the DRL and DLA values, indicating a probability that early root and shoot traits are regulated by separate genetic mechanisms. 

ANOVA followed by mean comparisons with the Tukey post-hoc test to compare DRL and DLA in the wheat genotypes of different genetic backgrounds (Figure 1) further verified the separate genetic mechanisms behind early root and shoot growth. Clear significant differences were noted for DRL among the genotype groups, with the significantly longest roots in the old Swedish breeding lines (8.50 cm), the wheat-rye introgression lines with chromosome 1R (11.78 cm), and 1RS (9.91 cm), as compared to the modern cultivars (4.20 cm) and genotypes with chromosome 2R (4.67 cm; Figure 1a). Differently, the significantly highest DLA was noted for the old lines (9.08 cm^2^) as compared to the genotypes with chromosome 1RS (6.37 cm^2^; Figure 1b) but no significant differences in DLA were found between the modern and old lines or among the introgression lines. Thus, the genes for early root vigour seemed to be present in the old Swedish breeding lines and on chromosome 1R and 1RS, while no presence of similar early vigour genes for shoot growth was indicated. 

The top 10% of genotypes (229, 216, 224, 227, 230, 219 and 221) for DRL all contained 1RS and showed values ranging between 12.84 cm and 16.13 cm (Figure 2a) while the top 10% of genotypes (197, 198, 200, 201, 222, 257 and 267) for DLA were spread among the genotype groups, i.e., old Swedish breeding lines and wheat-rye introgression lines with 1R, 2R and 3R, and the values ranged between 10.95 cm^2^ and 12.52 cm^2^ (Figure 2b).

### 2.2. Relationships between Drought Stresses and Plant Traits 

The ANOVA verified a significant effect of the drought treatments on all studied traits, i.e., root biomass (RB), days to heading (DTH), days to anthesis (DTA), tillers per plant (TPP), spikes per plant (SPP), productive spikes per plant (PSPP), flag-leaf area (FLA), spike length (SPL), 1,000-grain weight (TGW), grains per spike (GPS), grains per plant (GPP), grain weight per spike (GWPS), and grain weight per plant (GWPP) (Appendix A). A comparison of the mean values of the traits after drought stress at different growth stages (early and late) indicated a significant change in most of them due to the drought treatment (Appendix A). 

From the principal-component analyses (PCAs), with PC1 representing 39.5% and PC2 19.7% of the variation, the effects of the treatments were well differentiated along with the first principal component (PC1). Basically, all the measured traits were significantly and positively related with the control group (C) of plants (negative PC1), with a decrease in most traits with early drought stress (EDS) and an even larger decrease for late drought stress (LDS). PC2 generally differentiated the variation among the genotypes within a treatment for the evaluated traits (Figure 3).

### 2.3. Relationships among Investigated Traits 

The present study showed several significant correlations among the non-yield traits (FLA, DTH, DTA, TPP and RB) and yield traits (SPL, SPP, PSPP, TGW, GPP, GPS, GWPP and GWPS; Figure 4). In the non-stressed plants (C), the non-yield traits correlated mostly significantly and positively with GWPP and GPP, while negative correlations were mainly lacking between the non-yield and yield traits (Figure 4a). 

For the EDS plants, a higher number of significant positive correlations were found between non-yield (especially FLA and RB) and yield traits as compared to C plants (Figure 4b).

For the LDS plants, a higher number of significant negative correlations were found for DTH and DTA with the yield traits when compared to the C plants. In comparison, a lower number of significant positive correlations were found for TPP and RB with the yield traits (Figure 4c).

### 2.4. Genotypic Differences in Reactions to Drought Stresses

A large variation was obtained in plant performance among the genotypes after drought stress. As can be seen from Figure 5, the yield (GWPP) was found to mainly decrease due to EDS but even more so due to LDS, although a large and significant variation was obtained among the genotypes. The ranges of yield under C, EDS and LDS were from 2.02 g to 7.12 g, 1.54 g to 4.00 g and 0.08 g to 2.63 g, respectively which further verified the effects of the drought treatments presented by the mean comparison and PCA (Appendix A and Figure 3).

ANOVA followed by mean comparisons with the Tukey post-hoc test to compare the STI values of DTH, DTA, FLA, TPP, RB, SPP, PSPP, SPL, TGW, GPS, GWPS, GPP and GWPP revealed significant variations among the different genotype groups (modern, old, 1R, 1RS, 2R and 3R) under the two drought treatments (Appendix A). 

Under EDS treatment, basically, the modern genotypes significantly showed the highest STIs for GPS (0.95) and GWPS (0.96), while the genotypes with chromosome 1RS showed the generally lowest STIs of the yield traits (SPL: 0.80, GPS: 0.47 and GWPS: 0.54). Furthermore, a significantly contrasting performance in RB was noted between the genotypes with chromosome 3R (0.63) and the modern genotypes (0.14).

Under LDS treatment, higher STIs of yield traits were noted in the modern genotypes (SPP: 0.69, GPS: 0.75, GWPS: 0.54 and GWPP: 0.38) as compared to the other genotype groups, while the lowest yield performances were found in the genotypes with chromosome 1RS (SPL: 0.75, GPS: 0.35 and GPP: 0.21) and 3R (SPP: 0.36, TGW: 0.61 and GWPS: 0.24). A significantly contrasting performance in RB was noted between the genotypes with chromosome 3R (0.66) and the old genotypes (0.18).

Based on the STI values of GWPP, the genotypes 257 (3R), 256 (3R), 244 (3R), 227 (1RS), 202 (old), 208 (old) and 254 (3R) were found to be the top 10% genotypes under EDS with STI values ranging from 0.97 to 1.29, and the genotypes 238 (2R), 281 (modern), 204 (old), 273 (modern), 201 (old), 270 (2R) and 217 (1R) were found to be the top 10% genotypes under LDS with STI values ranging from 0.44 to 0.67 (Table 1). From the PCA with all the studied traits except GWPP, it was clear that different traits contributed to the high performance in terms of GWPP in the top 10% of genotypes. Under EDS, GPP and RB contributed to the highest extent to GWPP in the genotypes 254 (3R) and 256 (3R), while TGW and SPP had the highest impact on GWPP in the genotypes 227 (1RS) and 257 (3R), respectively (Figure 6a). Under LDS, TGW and FLA positively contributed to GWPP while GPP and RB showed a negative impact on GWPP in the genotypes 201 (old) and 217 (1R). On the contrary, a positive impact on GWPP was found with PSPP and RB in the genotype 238 (2R) (Figure 6b).

## 3. Discussion

Corresponding to the hypothesis of the present study, drought tolerance at various genotypes and genotype groups were investigated, representing different genetical constitutions. Thereby, the genetic background of the drought tolerance could be identified. Thus, old genotypes and alien wheat lines with 1R or 1RS showed a robust early root growth, which is most likely a characteristic of importance for drought-stress tolerance. However, the old genotypes and the lines with chromosome 1R and 1RS did not result in a high final-yield performance after drought stresses. Instead, the genotypes with 3R were found to be among the top 10% of genotypes that are tolerant to EDS while no clear pattern of the dominant genetic background was shown among the 10% best performing genotypes under LDS. Interestingly, separate genetic mechanisms for early root and shoot growth were demonstrated in the present study. Furthermore, our results demonstrated that RB, GPP and SPP were important characteristics that correlated with tolerance to EDS while TGW and FLA correlated positively with tolerance to LDS. This finding suggested that the general yield-based drought tolerance of wheat could be decomposed into multiple specific traits that potentially contribute to the high performance of the yield under drought.

A significantly higher early root growth, here measured as DRL, was found in the old wheat genotypes and alien genotypes with chromosome 1R and 1RS, as compared to modern wheat and wheat with other alien introgressions. Thus, genes contributing to early root growth seem to be present both in the wheat material consisting of old Swedish wheat lines and on chromosome 1RS. Previous studies have shown that an extensive root system in wheat enhances water and nitrogen capture, thereby explaining that a vigourous root system contributes to the yield performance of wheat under water deficiency [32]. An early root growth might be a trait of significant relevance for a good performance during drought stress, as early root growth might contribute to a higher chance of reaching the humidity during dry conditions [33]. A general understanding among Swedish growers is that old wheat varieties have a strong root growth, which contributes to their good performance during dry conditions, although scientific evidence for such a statement is lacking. However, in a recent study, a larger root system was observed in old wheat varieties than in modern wheat [34]. Furthermore, this large-to-small change in root biomass from old to modern wheat has been confirmed by several studies [35,36,37]. The relatively small root system in modern wheat could be attributed to the years of recurrent breeding programs that were mainly aimed at increasing the aboveground biomass, and especially the grain yield. Hence, the biomass shift from underground to aboveground was established along with years of yield-oriented selection. 

The findings of increased early root vigour in the genotypes with 1R and 1RS introgressions in the present study correspond well with the findings of other studies as reported below. Several studies have indicated positive performance at abiotic and biotic stress conditions by the introgression of 1RS in wheat [26]. The 1RS has been reported to confer disease resistance to wheat. For example, the resistance genes *Sr31*, *Yr9*, *Lr26* and *Pm8* were reported to be introduced to wheat along with chromosome 1RS, and effectively resisted or mitigated stem rust, stripe rust, leaf rust and powdery mildew, respectively [38,39]. A previous study using molecular markers provided evidence for the presence of *Sr31* in 1BL.1RS translocation lines [40]. Improved resistance to stripe rust and powdery mildew was reported in five 1BL.1RS translocation lines [41] and a newly detected gene, *Sr59* introgressed to wheat from 2R, has been reported to confer resistance to all currently known stem-rust races [29]. Translocations of 1RS to the long arm of wheat chromosome 1 (1AL, 1BL and 1DL) were shown to increase root biomass and yield in spring wheat [42]. The gene(s) for rooting ability have been suggested to be present in the distal 15% of the physical length of the 1RS arm [31]. Recently, several studies have provided evidence that the 1RS translocation improves the root traits of wheat [43,44,45]. In addition to the improvement of root traits, the 1RS translocation has been reported to contribute to increased grain yield [41,46,47]. Despite all the positive effects from introgression of the chromosome 1RS, grains of introgression lines containing 1RS were found to exhibit bread-making quality defects including sticky dough. The reduction in bread-making quality by 1RS has been related to the presence of the rye storage protein, secalin, which is encoded by *Sec-1*, and to a decrease in dough strength due to the loss of *Glu-B3/Gli-B1*-encoding gliadins and low-molecular-weight glutenins [48,49,50,51]. 

The results of the present study showed that both DRL and DLA varied significantly among genotypes, but a correlation between DRL and DLA was lacking. These findings indicate that early root and shoot growth of wheat are regulated by separate genetic mechanisms, which could possibly be either different genes or the same genes with different expression timings. However, previous studies have indicated a strong association between root and shoot traits at maturity [52,53,54], and old genotypes are reported to have both higher root and shoot biomass than modern genotypes [55,56]. Also, several dwarfing genes e.g., *Rht-B1c*, *Rht-D1c* and *Rht12* were found to significantly affect the root length of wheat at early stages [57]. Other studies have shown that a class of *Rht* genes did not affect shoot-growth traits such as coleoptile length, leaf-elongation rate and responsiveness to Gibberellin at early stages [58]. Another study showed that significant root-biomass differences observed at early vegetative stages among five wheat genotypes were clearly reflected in the leaf area and leaf biomass at later stages [59], which indicated the correlation between early root vigour and late shoot traits. Despite the above-mentioned findings, the relationships among root and shoot characteristics at early crop growth and the effects on the mature crop are still uncertain and need further evaluation.

The layout of the Biotron experiment under controlled conditions allowed us to selectively treat the plants with drought stress, explicitly avoiding other environmental variations, and to observe the variation of each of the 13 selected traits among the genotypes. Unexpectedly, the early root vigour of the old genotypes and the lines with 1RS did not contribute to the drought tolerance of these genotypes at maturity. We attribute the lack of correlation between early root vigour and drought stress to the fact that the experiment was carried out in pots in an indoor controlled-environment space, where a larger and more robust root system does not play a role in the opportunity to reach the humidity and nutritional elements. Field conditions provide considerably more space for the root system to expand in order to reach water and nutrient sources. From reported results based on wheat plants grown in a sufficiently large space, e.g., 1 m and 1.5 m PVC tubes [35] and 1.6 m columns [60], old wheat genotypes did show higher root biomass and longer roots than modern genotypes at maturity. Similarly, increased root biomass in wheat-rye introgression lines with chromosomes 1R and 1RS was noted by studies that were carried out in field conditions [43,44]. Long-rooting genotypes grown in conditions where the spread of their root systems have an impact on reaching water and nutrients should most likely be more drought tolerant than genotypes with smaller and less spread root systems. Studies in real field conditions are needed to evaluate the effect of the superior root system of old genotypes and the genotypes with chromosome 1R and 1RS on drought tolerance.

Our results suggest that 3R might be useful to improve early drought tolerance in wheat, which is an increasingly important characteristic for Nordic conditions, as well as under the predicted climate-change conditions. Differences in early drought tolerance were noted among the genotype groups evaluated here with the least change in yield (a high STI of GWPP) by the early drought treatment in the genotypes containing 3R. Additionally, four (257, 256, 244 and 254) of the top 10% of genotypes were identified as containing 3R. Previous studies on drought tolerance in wheat showed limited findings on stress-tolerance genes connected to chromosome 3R. To our knowledge, there is just one field study that proposes that genes regulating drought tolerance are present on chromosomes 7R, 3R and 5R [61]. Instead, 3R, along with 4R, 6R and 7R, has been well demonstrated as one of the rye chromosomes carrying major genes for the tolerance to aluminium toxicity [62,63,64,65,66]. Our results, indicating genes for drought tolerance on 3R, were achieved under EDS, differently to most other studies on drought tolerance in wheat-rye introgressions, which have mainly been based on drought stress imposed at late stages [12,42,67]. Thus, the timing of the drought stress might be the reason for the lack of data in the literature on drought-tolerance genes on 3R, and the potential tolerance genes on 3R might be specific to early drought. The PCA result based on the top 10% of genotypes for EDS suggested that a large root system, together with high numbers of grains and spikes per plant as beneficial traits for high tolerance to early drought. Thus, the 3R chromosome seemed to contribute to the ability for the plant to set roots and spikes regardless of whether there had been an early drought period, which is a characteristic that might be of importance for the future climate change in the Nordic countries, where early drought seems to be an increasingly common feature.

The present study clearly indicated different mechanisms of wheat to combat to drought stress at different maturation stages. This was verified by the fact that the genotype groups with early root vigour and EDS tolerance were clearly distinguishable, while no clear pattern of LDS tolerance in a specific genotype group was detected in the present study. As anthesis is a critical period for grain formation in wheat, with a high impact on meiosis from drought [68], large impacts on final yield have often been described as a result of late drought [8,69]. The potential of the grain weight of wheat at maturity is closely determined by the number of endosperm cells per grain and the process of cell division in the endosperm, which ends within two to three weeks after anthesis [70]. Another factor determining the grain weight is cell expansion, which was found to be related to water content during the grain-filling period [71]. Drought stress at anthesis has been reported to shorten the grain-filling period of wheat [72], and the termination of cell expansion predetermines the maturation of grains [73,74]. Thus, the water deficit during this period is known to cause a severe reduction in grain yield. Unlike the direct and severe effects of post-anthesis drought on the final yield, the effects of early drought at the vegetative growth stages are more morphology-related, e.g., EDS affects plant height, leaf area and number of tillers [75]. In the present study, the PCA results based on the top 10% of genotypes for LDS indicated that TGW and FLA might account for the high tolerance of genotypes 201 (old) and 217 (1R), while RB and GPP showed a negative correlation to the high tolerance. Hence, large grain size and flag leaves are beneficial traits for high tolerance to late drought, and the negative contribution of RB and GPP to the high tolerance to late drought further confirmed the different mechanisms of wheat plants to combat early and late drought.

Yield performance under drought stress is the major concern when breeders develop drought-tolerant crop varieties [76] because, in a broad sense, drought tolerance is defined as the yield-maintaining ability of a plant under drought. The empirical approach used in breeding programs for drought-tolerant genotypes emphasizes the yield under both non-stressed and stressed conditions, so that both yield stability and yield potential can be evaluated [77]. However, the complexity of drought tolerance has been widely accepted, particularly from a genomics perspective, due to the identification of many drought-responsive genes [78]. The *Lea* genes encoding late-embryogenesis-abundant (LEA) proteins were reported to be highly related to drought tolerance [79,80,81]. Another large drought-related gene family is the *NAC* genes that encode transcription factors. By introducing a rice *NAC1* gene (*SNAC1*) into a wheat variety, significantly enhanced drought tolerance was achieved in the transgenic wheat plants [82]. The V-PPase gene family that encodes an enzyme vacuolar H+-translocating pyrophosphatase was found to be responsive to abiotic stress. Improved drought tolerance was observed in transgenic *Arabidopsis* plants that were overexpressing a V-PPase member of wheat *TaVP1* [83]. Therefore, due to the polygenic nature of drought tolerance, using yield as the only selection criterion to screen for a drought-tolerant genotype might not be the optimal approach. Valuable information on drought tolerance that is hidden behind other traits could be missed. However, blindly including too many traits into a selection process could result in an unnecessary increase in workload and provide a misleading result. As demonstrated earlier in this paper, we propose that the yield-based drought tolerance of wheat can be explained by other traits, and therefore identifying and combining the most yield-contributing traits during selection may help achieve more effective results than selecting based on yield alone.

## 4. Materials and Methods

### 4.1. Plant Materials

A collection of 73 spring-wheat (*Triticum aestivum*) genotypes consisting of 14 Swedish landraces and old cultivars released from 1928 to 1990 [84], 50 wheat-alien introgression lines [85] with rye chromosomes 1R, 2R, 3R, 4R, 5R, 6R and Leymus Racemosus chromosome N in the form of translocation and substitution [29,40], and nine modern wheat breeding lines from the company Lantmännen were used in this study (Appendix A). All the wheat lines with different genetic backgrounds (old, modern, introgression lines) will be referred to as genotypes from here and onwards in this publication.

### 4.2. Early Root and Shoot Phenotyping

To evaluate the early root growth of the genotypes, a hydroponic experiment was conducted in the greenhouse, germinating the wheat seeds on wet filter paper at low-temperature conditions (4 °C, 48 h) for uniform germination. Following previously described methods [13,86], germinated seeds were fixed on dedicated blue blotter paper (210 mm × 297 mm, Anchor paper company) with small clips and covered by black plastics. The wetness of the paper was maintained by dipping its bottom in water. After seven days of growth under room temperature (25 °C/18 °C day/night), root imaging was performed with a DSLR camera (Canon 1300D, Canon Inc., Tokyo, Japan) mounted on a kaiser stand 40 cm above the root surface. The image-based digital-root length (DRL) was extracted using the software RootNav [87].

To evaluate the early shoot growth, the genotypes were phenotyped in the Biotron (growing conditions described below) from three different angles using two digital single-lens reflex (DSLR) cameras (Canon 1300D, Canon, USA) 20 days after sowing as previously described [88]. Cameras were operated through the software digiCamControl (http://digicamcontrol.com/, accessed on 28 February. 2020). The image-based digital-leaf area (DLA) was extracted using the software Easy Leaf Area (https://www.quantitative-plant.org/software/easy-leaf-area, accessed on 28 April 2020).

Comparisons of DLA and DRL were made on an individual-genotype basis, but groups of genotypes were also compared (i.e., modern wheat genotypes, old Swedish genotypes, 1R wheat-alien introgression lines, 1BL.1RS wheat-alien translocation lines, 2R wheat-alien introgression lines, 3R wheat-alien introgression lines).

### 4.3. Biotron Trial

The experiment was carried out in 2020, growing the genotypes from April–September in a daylight (DK) chamber with natural light in the Biotron at the Swedish University of Agricultural Sciences in Alnarp, Sweden. The temperature and humidity were strictly regulated hourly based on the mean weather data of Malmö over the past decade (2010–2019). The data were obtained from Swedish Meteorological and Hydrological Institute (SMHI) (Appendix A). Five seeds of each genotype and for each treatment were sown in each 2.5 L plastic pot filled with peat-based soil, and after thinning, three plants were retained in each pot. Thus, three biological replicates of each genotype were subjected to each of the three growing conditions, i.e., standard growing conditions used as a control (C), early-drought-stress (EDS) growing conditions, and late-drought-stress (LDS) growing conditions. To achieve uniform solar irradiance for growth throughout the experiment, the position of each pot was shifted within the chamber weekly.

### 4.4. Growing Conditions including Drought Stress

For the C treatment, the plants were watered every second day so that the soil moisture was maintained throughout the whole cultivation period and watering was stopped when the spikes turned yellowish in order for the plants to become mature. The EDS-treated plants were watered similarly to the C plants until day 30 (Zadoks 23) after planting [89] when the drought treatment started, by withholding water for 28 days, and thereafter the watering again followed the C treatment. Similarly, the LDS plants followed the C treatment until day 60 (beginning of heading stage, Zadoks 50) after planting, when water was withheld for 14 days, and thereafter the C treatment started again. The soil moisture on the last day of both drought treatments was below 1%, as measured by a moisture meter (Takemura electric works DM-15 soil PH & moisture tester hygrometer).

### 4.5. Morphological, Phenological and Yield Traits

Days to heading (DTH) (number of days taken from sowing to appearance of spikes) and anthesis (DTA) (number of days taken from sowing to appearance of anthers) were recorded manually. Spike length (SPL) was measured with a ruler in centimetres. Tillers per plant (TPP), spikes per plant (SPP), productive spikes per plant (PSPP), grains per plant (GPP) and grains per spike (GPS) were counted. Flag-leaf area (FLA) [90] was measured with an LI-3000C Portable Leaf Area Meter (LI-COR Environmental). Grain weight per plant (GWPP), grain weight per spike (GWPS), 1000-grain weight (TGW) (CONTADOR seed counter, PFEUFFER, Kitzingen, Germany) and dry root biomass (RB) were measured in grams. Thus, yield per plant = GWPP.

### 4.6. Data Analysis

The stress-tolerance index of five yield traits, including GWPP, GWPS, GPP, GPS, and TGW, was calculated to determine the comprehensive responses of genotypes to drought stresses by using the following formula.

Stress-tolerance index (STI) [15]:STI = (Ys × Yp)/(Yp)^2^,(1)
where Ys represents the yield trait of each genotype under drought-stress conditions; Yp and Yp represent the yield trait of each genotype and the mean yield trait of all genotypes in the productive conditions, respectively.

All the statistical analyses were performed using software RStudio [91], version 1.2.5042. A two-way ANOVA was conducted for each trait to detect significant differences between the treatments. Where significant differences (*p* < 0.05) were detected, multiple mean comparisons were conducted using the Tukey post-hoc test with the package rstatix. Pearson’s correlation coefficients were calculated to investigate correlations among different traits with their mean values using the package Hmisc, and the results were visualized using the package corrplot. Principal-component analysis (PCA) was performed with the package ggfortify to further explore the relationship among different drought-related traits and treatments.

## 5. Conclusions

Global food security is currently threatened by dynamic climate changes and the growing global population, leading to an increase both in abiotic stress conditions and in demands. To cope with this situation, and with the loss of genetic diversity in modern breeding lines over the last century, diverse genetic materials are untapped resources with which to search for candidate genes that contribute to high yield under stress conditions. Early root vigour is a useful characteristic for the plant to sustain stressful growing conditions, e.g., from drought, which is present in the old Swedish lines and lines with 1R and 1RS. The early root vigour in such lines is not necessarily correlated to early shoot growth, indicating the possibility of different genetic determinants of these characteristics. The 3R may contain important genes contributing to tolerance in wheat for early drought stress, which is a characteristic already of importance under Nordic conditions, although early drought tolerance is expected to become even more important with the predicted climate change. The mapping of the responsible genes and the plant traits determining early drought tolerance is therefore an important challenge. Here, grains per plant, root biomass and spikes per plant were the dominating traits that contributed to a low change in yield in the genotypes with good tolerance to early drought. This indicated the ability to grow and set seeds despite an early drought period as an important property of early drought tolerance. Differently, the 1000-grain weight and a large flag-leaf area were the most important traits for a low change in yield and therefore a good tolerance to late drought. Thus, breeding targets related to early vigour or breeding for drought resistance at various developmental stages need to be set, and suitable genes need to be determined and mapped in order to successfully breed drought-tolerant spring wheat.

## Figures and Tables

**Figure 1 ijms-23-03333-f001:**
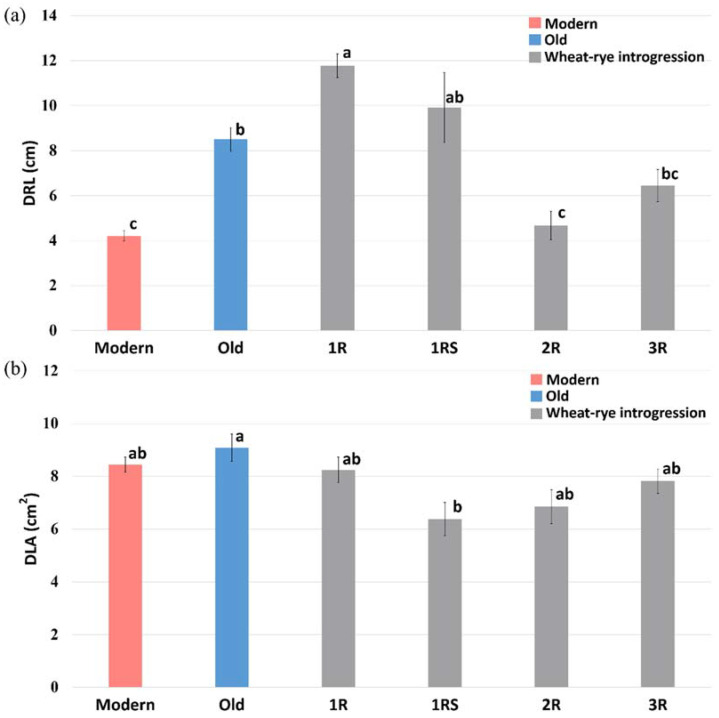
Comparisons of early crop vigour measured as (**a**) digital-root length (DRL) and (**b**) digital-leaf area (DLA) in spring-wheat lines of different genetic background, including modern lines, old Swedish lines and wheat-rye introgression lines with different chromosomes from rye. The results of Tukey post-hoc test are presented by applying a compact letter display at *p* < 0.05. Modern = approved cultivars and breeding lines received from company Lantmännen, Old = cultivars released from 1928 to 1990, 1R = introgressions of chromosome 1R, 1RS = introgressions of chromosome 1RS, 2R = introgressions of chromosome 2R, 3R = introgressions of chromosome 3R.

**Figure 2 ijms-23-03333-f002:**
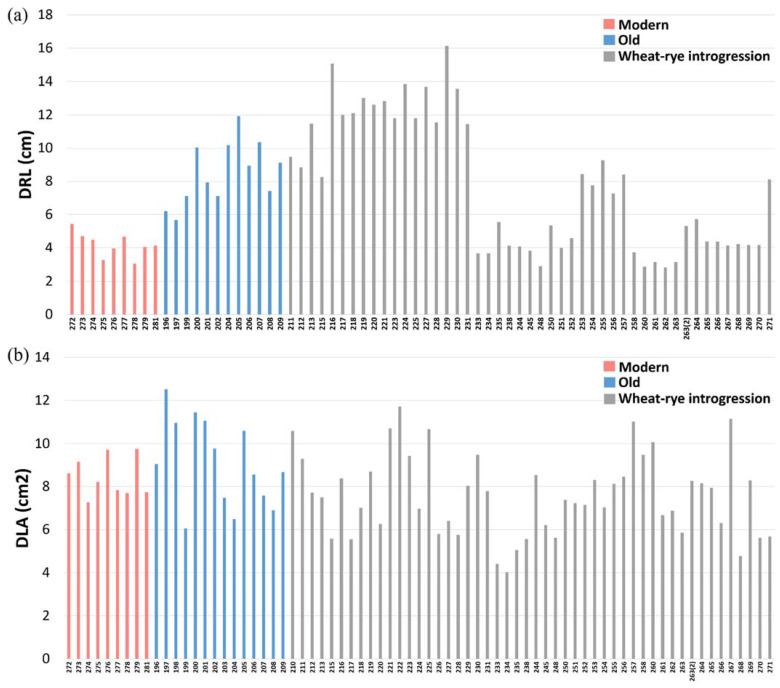
Comparisons of early crop vigour measured as (**a**) digital-root length (DRL) and (**b**) digital-leaf area (DLA) among genotypes including modern lines (approved cultivars and breeding lines received from company Lantmännen), old Swedish lines (cultivars released from 1928 to 1990) and wheat-alien introgression lines (with 1R, 1RS, 2R and 3R). Due to the unstable germination power, five genotypes (198, 203, 210, 222, 226) were missing from the DLA data.

**Figure 3 ijms-23-03333-f003:**
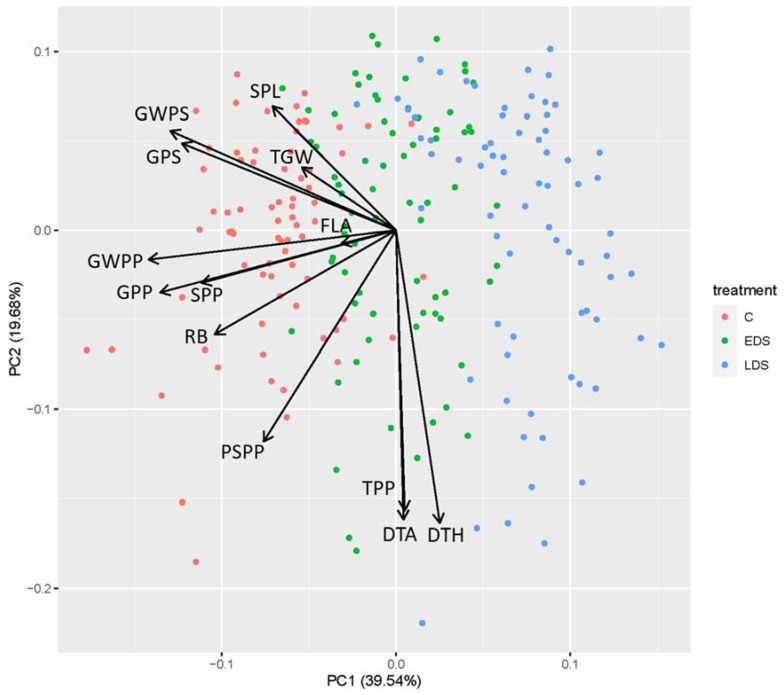
Biplot from principal-component analysis (PCA) for traits, root biomass (RB), days to heading (DTH), days to anthesis (DTA), tillers per plant (TPP), spikes per plant (SPP), productive spikes per plant (PSPP), flag-leaf area (FLA), spike length (SPL), 1000-grain weight (TGW), grains per spike (GPS), grains per plant (GPP), grain weight per spike (GWPS), and grain weight per plant (GWPP) in 73 spring-wheat genotypes under control (C), early drought stress (EDS) and late drought stress (LDS).

**Figure 4 ijms-23-03333-f004:**
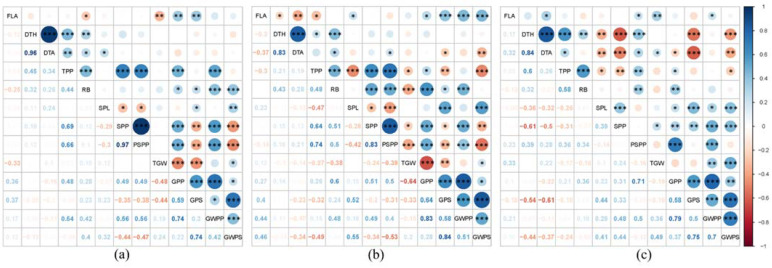
Correlation among morphological, phenological and yield traits flag-leaf area (FLA), days to heading (DTH), days to anthesis (DTA), tillers per plant (TPP), root biomass (RB), spike length (SPL), spikes per plant (SPP), productive spikes per plant (PSPP), 1,000-grain weight (TGW), grains per plant (GPP), grains per spike (GPS), grain weight per plant (GWPP), grain weight per spike (GWPS) studied in wheat under (**a**) control (C), (**b**) early drought stress (EDS) and (**c**) late drought stress (LDS). ***: sig. < 0.001, **: sig. < 0.01, *: sig. < 0.05.

**Figure 5 ijms-23-03333-f005:**
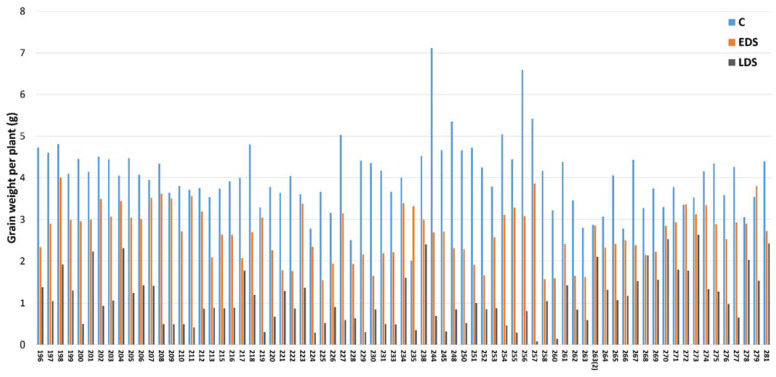
Yield (grain weight per plant **=** GWPP) of each genotype under control (C), early drought stress (EDS) and late drought stress (LDS).

**Figure 6 ijms-23-03333-f006:**
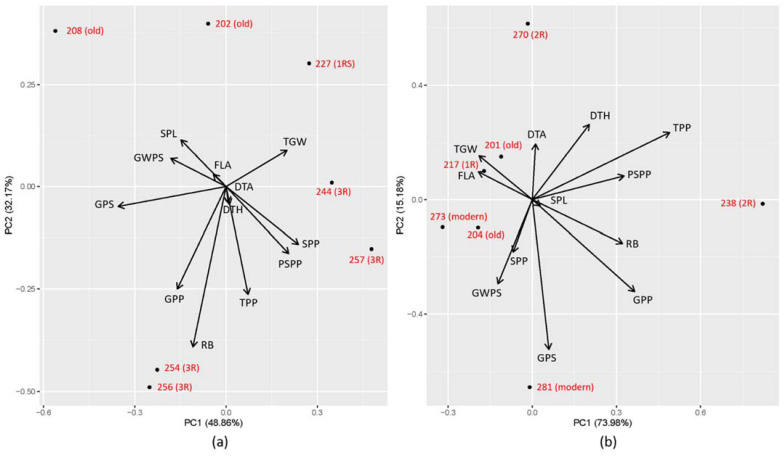
Principal-component analysis (PCA) for stress-tolerance index (STI) of all traits except grain weight per plant (GWPP) with the top 10% genotypes selected by stress-tolerance index (STI) of GWPP under (**a**) early drought stress (EDS) and (**b**) late drought stress (LDS). Modern = approved cultivars and breeding lines received from company Lantmännen, Old = cultivars released from 1928 to 1990, 1R = introgressions of chromosome 1R, 1RS = introgressions of chromosome 1RS, 2R = introgressions of chromosome 2R, 3R = introgressions of chromosome 3R.

**Table 1 ijms-23-03333-t001:** Top 10% genotypes selected by stress-tolerance index (STI) of yield (grain weight per plant = GWPP) under early drought stress (EDS) and late drought stress (LDS).

EDS			LDS		
Genotype	STI	Genetic Background	Genotype	STI	Genetic Background
257	1.29	3R	238	0.67	2R
256	1.25	3R	281	0.66	modern
244	1.18	3R	204	0.58	old
227	0.98	1RS	273	0.57	modern
202	0.97	old	201	0.57	old
208	0.97	old	270	0.51	2R
254	0.97	3R	217	0.44	1R

## Data Availability

The data is openly available with the first author and can be accessed on reasonable request.

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
