# Peer review of "Climate Change Impact on Wheat Performance—Effects on Vigour, Plant Traits and Yield from Early and Late Drought Stress in Diverse Lines"

_ijms, 2022, doi:10.3390/ijms23063333_

Round 1

Reviewer 1 Report

Dear Authors

the aim of the present study was to use the performance of a broad set of phenotypic traits to characterize drought stress tolerance in a wide material of modern, ancient, old, and alien introgressed spring wheat lines,  subjecting plants to early or late drought stress in controlled conditions.
The performance of plants was evaluated at the seedling and maturity stage using a combination of classical agronomic traits, including the calculation of STI, and an image-based phenotyping technique.

The work is well done. The manuscript is clear and interesting. I just added a few easy-to-make fixes.

Lines 32-34 Add the following sentence

With the pronounced global climate changes (i.e. rising temperature), prolonged shortages of water supply (drought stress) are becoming increasingly frequent depleting ecophysiological performance of plants [2-3].

[2] Cataldo, E., Salvi, L., & Mattii, G. B. (2021). Effects of irrigation on ecophysiology, sugar content and thiol precursors (3-S-cysteinylhexan-1-ol and 3-S-glutathionylhexan-1-ol) on Vitis vinifera cv. Sauvignon Blanc. Plant Physiology and Biochemistry, 164, 247-259.

[3] Cataldo, E., Salvi, L., Paoli, F., Fucile, M., & Mattii, G. B. (2021). Effect of Agronomic Techniques on Aroma Composition of White Grapevines: A Review. Agronomy, 11(10), 2027.

Line 365 Triticum aestivum

Best regards,

Author Response

Dear Reviewer,

We appreciate the valuable comments and suggestions, which we have tried to constructively use to improve the quality of our manuscript. The manuscript has now been revised following every point of the reviewers suggestions. Changes in the manuscript were made according to reviewers’ comments and are addressed point by point as below:

The aim of the present study was to use the performance of a broad set of phenotypic traits to characterize drought stress tolerance in a wide material of modern, ancient, old, and alien introgressed spring wheat lines,  subjecting plants to early or late drought stress in controlled conditions.
The performance of plants was evaluated at the seedling and maturity stage using a combination of classical agronomic traits, including the calculation of STI, and an image-based phenotyping technique.

The work is well done. The manuscript is clear and interesting. I just added a few easy-to-make fixes.

Thanks for the positive comments.

Lines 32-34 Add the following sentence

With the pronounced global climate changes (i.e. rising temperature), prolonged shortages of water supply (drought stress) are becoming increasingly frequent depleting ecophysiological performance of plants [2-3].

[2] Cataldo, E., Salvi, L., & Mattii, G. B. (2021). Effects of irrigation on ecophysiology, sugar content and thiol precursors (3-S-cysteinylhexan-1-ol and 3-S-glutathionylhexan-1-ol) on Vitis vinifera cv. Sauvignon Blanc. Plant Physiology and Biochemistry164, 247-259.

[3] Cataldo, E., Salvi, L., Paoli, F., Fucile, M., & Mattii, G. B. (2021). Effect of Agronomic Techniques on Aroma Composition of White Grapevines: A Review. Agronomy, 11(10), 2027.

The suggested sentence and new references have been added accordingly.

Line 365 Triticum aestivum

Thanks for pointing out this. The format correction has been made.

As can be seen from the point by point answers to reviewer comments, we have made every attempt to improve the manuscript following suggestions from the reviewers. We thereby hope that the manuscript is now suitable for publication in International Journal of Molecular Science.

Reviewer 2 Report

The authors aimed to determine the effects of drought stress and the drought stress tolerance in different spring wheat (modern, old, wheat-rye introgressions) lines.

Please follow the journal’s criteria and instructions in the Abstract, Discussion chapters and by the references in the text.

Please take into consideration to change “used” to “examined” or “tested” in L11.

Please explain the meaning of “evaluating to 10% genotypes” in L16.

The authors should make attention to the minor lingual mistakes (e.g. vigor vs. vigour, “area harvested”, scientific names of plants, etc.).

In L46-47 please take into consideration to rephrase “are not directly correlated to yield” to “many traits beside yield”.

The P-value in the text (L85) and the caption of Figure 1. are not the same.

In the captions of the figures please explain all the abbreviations and please take into consideration to name “modern” and “old” genotype as well.

This sentence: “ANOVA clearly showed that both digital root length (DRL) and digital leaf area (DLA) varied significantly” is not quite right according to Figure 1.

There I could not find Table S1-S5 in this manuscript but maybe they are in another version of the manuscript or in a file that I did not receive yet. In my point of view, these tables are necessary for a manuscript.

Please determine the exact developmental stage of plants at the beginning of the stress treatment with a uniform scale, like BBCH or Zadocks.

Please explain the meaning of abbreviations by the first appearance in every chapter. 

According to the journal in the Discussion chapter, “authors should discuss the results and how they can be interpreted in perspective of previous studies and of the working hypotheses”. In the current form this chapter is not comparative enough, please rewrite or rephrase this chapter according to the criteria.

Maybe in Author Contributions, the initials of the authors’ names will be enough.

By the References, please make attention to the journal’s criteria.

Author Response

Dear Reviewer,

We appreciate the valuable comments and suggestions, which we have tried to constructively use to improve the quality of our manuscript. The manuscript has now been revised following every point of the reviewers suggestions. Changes in the manuscript were made according to reviewers’ comments and are addressed point by point as below:

The authors aimed to determine the effects of drought stress and the drought stress tolerance in different spring wheat (modern, old, wheat-rye introgressions) lines.

Please follow the journal’s criteria and instructions in the Abstract, Discussion chapters and by the references in the text.

Please take into consideration to change “used” to “examined” or “tested” in L11.

Thanks for the suggestion, “used” has been changed to “examined” as suggested.

Please explain the meaning of “evaluating to 10% genotypes” in L16.

Thanks for pointing out this, the sentence has been corrected accordingly.

The authors should make attention to the minor lingual mistakes (e.g. vigor vs. vigour, “area harvested”, scientific names of plants, etc.).

Thanks for the comment. Lingual mistakes have been corrected accordingly.

In L46-47 please take into consideration to rephrase “are not directly correlated to yield” to “many traits beside yield”.

Thanks for the comment. The sentence has been changed accordingly.

The P-value in the text (L85) and the caption of Figure 1. are not the same.

The P-value has now been corrected to 0.001. It describes the significance of the variation among genotypes (Table S1), not among genotype groups (Figure 1), so the “(Table S1)” has been inserted here.

In the captions of the figures please explain all the abbreviations and please take into consideration to name “modern” and “old” genotype as well.

Thanks. The suggested improvements of the figures have been included.

This sentence: “ANOVA clearly showed that both digital root length (DRL) and digital leaf area (DLA) varied significantly” is not quite right according to Figure 1.

This sentence describes variation among genotypes (Table S1), not among genotype groups (Figure 1). The “(Table S1)” has been inserted here.

There I could not find Table S1-S5 in this manuscript but maybe they are in another version of the manuscript or in a file that I did not receive yet. In my point of view, these tables are necessary for a manuscript.

All the supplementary tables are now added to the compressed file and will be resubmitted along with the revised manuscript.

Please determine the exact developmental stage of plants at the beginning of the stress treatment with a uniform scale, like BBCH or Zadoks.

Thanks for the suggestion. In Materials and Methods, the growth stages of plants at the beginning of treatments are now marked in the form of Zadoks growth scale as suggested.

Please explain the meaning of abbreviations by the first appearance in every chapter. 

Thanks for this comment. We have now followed the journal instructions. Thus, the abbreviations have been defined at the first appearance in each of the sections: abstract, main text, the first figure or table.

According to the journal in the Discussion chapter, “authors should discuss the results and how they can be interpreted in perspective of previous studies and of the working hypotheses”. In the current form this chapter is not comparative enough, please rewrite or rephrase this chapter according to the criteria.

Thanks for the suggestion. A hypothesis has been added to the introduction and the discussion has been revised along the suggested comparative style.

Maybe in Author Contributions, the initials of the authors’ names will be enough.

All the names in Author Contributions have been changed into the form of initials as suggested.

As can be seen from the point by point answers to reviewer comments, we have made every attempt to improve the manuscript following suggestions from the reviewers. We thereby hope that the manuscript is now suitable for publication in International Journal of Molecular Science.

Round 2

Reviewer 2 Report

The authors of this manuscript aimed to determine the effects of drought stress in a wide material of modern, old and alien introgressed spring wheat lines.

Please clarify all the abbreviations in the footer or title of the figures and tables.

Please add a reference to the mentioned Zadoks scale.

Author Response

Dear Reviewer,

We appreciate the valuable comments and suggestions, which we have tried to constructively use to improve the quality of our manuscript. Changes have now been made to the manuscript accordingly and your comments are addressed point by point as below:

The authors of this manuscript aimed to determine the effects of drought stress in a wide material of modern, old and alien introgressed spring wheat lines.

Please clarify all the abbreviations in the footer or title of the figures and tables.

Thank you for the suggestion. Corresponding clarifications of abbreviations are now added to all the figures and tables.

Please add a reference to the mentioned Zadoks scale.

Thank you for the suggestion. A reference to the Zadoks scale has now been added.

Thank you again for your efforts. We have incorporated your suggestions to improve this manuscript. We thereby hope that the manuscript is now suitable for publication in the International Journal of Molecular Science.
